# HEARING FACES AMONG HOMOGENEOUS POPULATIONS: IMPROVEMENT OF CROSS-MODAL BIOMETRICS

## ABSTRACT

The relationship between voice and face is well-established in neuroscience and biology. Recent algorithmic advancements have yielded substantial improvements in voice face matching. However, these approaches predominantly achieve success by leveraging datasets with diverse demographic characteristics, which inherently provide greater inter-speaker variability. We address the challenging problem of voice face matching and retrieval in homogeneous datasets, where speakers share gender and ethnicity. Our novel deep architecture, featuring a weighted triplet loss function based on face distances, achieves state-of-the-art performance for voice face matching on these uniform populations. We evaluate our model on a sequence of homogeneous datasets containing only voices and faces of people sharing gender and ethnic group. In addition, we introduce percentile-recall, a new metric for evaluating voice face retrieval tasks.

## 1 INTRODUCTION

Neuroscientists and biologists have long established a strong correlation between human appearances and voices (Mavica & Barenholtz, 2013; Smith et al., 2016a;b). As Wells et al. (2013) demonstrates, genetic information and hormone levels during puberty shape both voice-controlling organs and facial features. The strength of this phenomenon is evident in mundane interactions: during phone calls, we can often deduce various demographic details about the person with whom we talk, such as their gender, ethnicity, and approximate age. Conversely, watching muted TV shows, we may be able to reconstruct characters' voices, at least to some extent.

Recent advances in deep learning, particularly in face (Kim et al., 2022; 2024) and audio recognition (Koluguri et al., 2022; Chen et al., 2022a;b), allow studying this link from a precise algorithmic perspective. Nagrani et al. (2018a) were the first to explore the *voice face matching* problem (see Figure 1a) as well as face voice matching, proposing a CNN-based deep neural network to embed both modalities into a common latent space. The authors introduced a metric for the task called *identification accuracy* (which we recall in subsection 2.3.1). This metric involves matching a voice sample to a correct face out of two choices (or in face voice matching, a face to

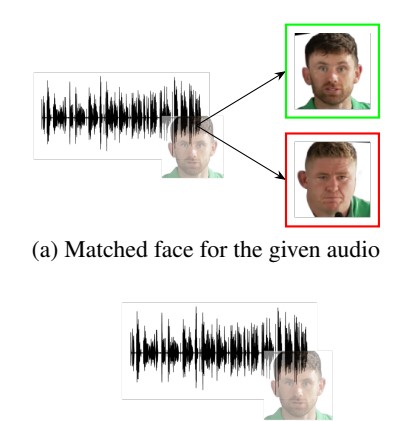

(a) Matched face for the given audio

(b) Retrieved (Sorted) gallery

Figure 1: (a) Voice face matching: associating a given audio sample the corresponding face out of several possibilities. (b) Voice face retrieval: sorting a given gallery of faces by proximity to a given voice sample.

| Gender | Demographic Group | Train | | | Val | | | Test | | |
|---|---|---|---|---|---|---|---|---|---|---|
| | | # IDs | # Faces | # Speech | # IDs | # Faces | # Speech | # IDs | # Faces | # Speech |
| Male | Asian | 16893 | 113868 | 200533 | 2137 | 14173 | 14116 | 2103 | 14169 | 15205 |
| | Latino-Hispanic | 10646 | 84065 | 115509 | 1381 | 10876 | 13430 | 1408 | 11068 | 12757 |
| | Middle-Eastern | 8389 | 63269 | 88096 | 1026 | 7851 | 10971 | 1070 | 7994 | 10476 |
| | White | 52111 | 390959 | 464928 | 6558 | 49292 | 61456 | 6470 | 48498 | 58075 |
| Female | Asian | 4802 | 32377 | 32615 | 567 | 3748 | 2907 | 579 | 4024 | 4557 |
| | Latino-Hispanic | 3105 | 22492 | 24571 | 359 | 2615 | 2950 | 394 | 2879 | 2806 |
| | Middle-Eastern | 1239 | 8405 | 10079 | 149 | 989 | 1328 | 160 | 1099 | 1763 |
| | White | 21852 | 162648 | 209712 | 2773 | 20821 | 23584 | 2678 | 20066 | 22723 |

Table 1: Dataset statistics showing the number of identities, face images, and speech segments for each demographic group across train/val/test splits.

correct audio out of two choices). The authors provided a human baseline for the voice face matching task and demonstrated that their system's performance is nearly on par with human ability.

Building upon this foundation, subsequent works have further explored voice to face (and face to voice) matching. Nagrani et al. (2018b) exhibited similar performance while introducing a curriculum learning schedule for hard negative mining. Wen et al. (2019) were the first to surpass human-level performance with their DIMNet: a deep architecture learning common representations for faces and voices through their relationships to demographic covariates such as gender and nationality. They achieved an accuracy of 84.12% in the identification metric. Recently, Zhu et al. (2022) proposed a more accurate system, achieving 85.3% by combining contrastive learning with unsupervised techniques.

Nevertheless, previous works attain these scores in the identification accuracy by considering heterogeneous datasets, including speakers of various genders and ethnicities, thus inducing larger variance in the vocal and facial feature spaces. As Nagrani et al. (2018a) note, human performance deteriorates markedly when assessed on voice face matching of speakers sharing gender, ethnicity, or age group. Similarly, the works mentioned above, akin to human capabilities, fall short in distinguishing between speakers sharing one or more of these covariates.

We propose a new deep architecture for studying common latent representations of voices and faces from "homogeneous" datasets. Our system is based on a weighted triplet loss, where the weights are a function of the distance between faces (or rather their embeddings under a face encoder). This particular choice of loss allows us to identify speakers among people sharing their gender and ethnic group.

Another formulation of the voice face problem is *voice face retrieval* (see Figure 1b): given an audio sample of a speaker and a face gallery containing one or more face images of the person to whom the recording belongs. We then compute how similar the correct face(s) are to the given audio and sort this gallery by the distance of the faces from the given audio. To measure the performance of our deep architecture in this task, we propose a new metric, *percentile recall*, which is highly correlated with identification accuracy. This metric allows us to quantify the quality of sorting mechanism for galleries obtained in response to a given voice "query".

ORGANIZATION OF THE PAPER

We dedicate section 2 for introducing the implementation details. section 3 is dedicated for describing the model, section 4 for the results and section 5 for conclusions. We provide in the appendices several insights about the influence of codecs and noising, separately and together on our model.

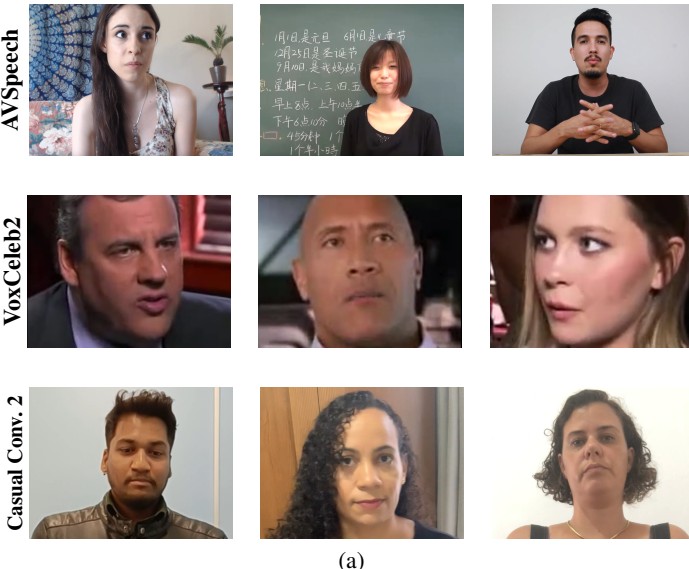

(a)

Figure 2: Samples from the heterogenic datasets used for pre-training our model. From top to bottom, samples from the datasets AVSpeech, VoxCeleb2 and Casual Conversations 2. Through a pre-training session the model learns to some extent association of voice to face. Splitting AVSpeech, we obtain plethora of homogeneous datasets.

## 2 IMPLEMENTATION DETAILS

### 2.1 DATASETS

#### 2.1.1 HETEROGENEOUS DATA

We performed pre-training on the several homogeneous datasets, that is with male and female speakers of various ethnicities,

- VoxCeleb2 (Nagrani et al., 2019): a dataset containing over 1 million clips corresponding to 6,112 celebrities, mostly white, The data is extracted from 150,000 YouTube videos of total length 2442 hours.
- Casual conversations 2 (Porgali et al., 2023): a dataset collected by Meta including 26,467 videos of 5,567 unique paid participants, with an average of almost 5 videos per person. People appearing in the dataset originate from Asia, South and North America, representing diverse demographic characteristics.
- AVSpeech(Ephrat et al., 2018): a large-scale dataset comprising speech video clips with no interfering background noises. Each YouTube video was divided to short clips of varying lengths where the audible sound in the soundtrack is assumed to belong for a single speaking person, visible in the video. The dataset contains 4700 hours of video segments, from a total of 290k YouTube videos, spanning a wide variety of people, languages and face poses. After pre-processing the clips we were able to extract approximately 150,000 different identities.

We supply in Figure 2 several face samples from each of the three datasets.

#### 2.1.2 HOMOGENEOUS DATASETS

We partitioned the AVSpeech dataset, being a diverse collection of audiovisual data, previously characterized in Oh et al. (2019), into distinct homogeneous subsets based on gender and ethnicity demographics. To determine individual demographic attributes, we employed the DeepFace framework Serengil & Ozpinar (2021). The detailed composition of each homogeneous subset is presented in Figure 2. To maintain experimental integrity and since we perform fine-tuning of a model trained

on the heterogeneous data above, we preserved the original train-validation-test splits when creating these specialized subsets, thereby preventing data leakage across partitions. Specifically, subjects appearing in any particular split (training, validation, or test) of the complete AVSpeech dataset were consistently allocated to the corresponding split in their respective homogeneous subsets. Due to limited representation of Black and Indian populations in the DeepFace-classified data, we determined that training voice-face models for these demographic groups would not yield statistically meaningful results at this time.

## 2.2 PRE-PROCESSING

In order to extract face images and corresponding audio samples from videos we first apply an active speaker detector to split the video to smaller clips with a single speaker. Then, on each of these clips we performed

- audio operations: split the audio into short segments ranging between 3 to 10 seconds.
- image operations: we first filter faces of bad quality; those having yaw, pitch or roll greater than $15°$, with closed eyes or with open mouths. We then align the face using MTCNN landmark extractor (Zhang et al., 2016), remove background and resize the image to be of size $256 \times 256$.

In order to generate the final image and audio datasets we clustered audios and images by using DBSCAN on the Hadamard product of the matrices indicating the cosine distance between each pair of embeddings. Altogether our procedure yields for each speaker several facial images, vocal samples and corresponding embeddings.

### NOTATIONS

Henceforth we denote the vector latent representation of a voice by $\boldsymbol{v} \in \mathbb{R}^L$ and the vector latent representation of a face by $\boldsymbol{f} \in \mathbb{R}^L$. Enumerating speakers in the dataset from 1 to $N$ we denote an audio vector representation of the $j-$th speaker by $\boldsymbol{v}_j \in \mathbb{R}^L$ and a face vector representation of them by $\boldsymbol{f}_j \in \mathbb{R}^L$.

## 2.3 METRICS

### 2.3.1 IDENTIFICATION AND BINARY ACCURACY

The performance of voice face matching architectures is often evaluated in *the identification accuracy* (also known as 1:2-metric). Fixing $N'$ triplets of the form $(\boldsymbol{v}_{i_1}, \boldsymbol{f}_{\sigma(i_1)}, \boldsymbol{f}_{\sigma(i_2)})$ with $i_1 \neq i_2 \in \{1, \ldots, N'\}$ and $\sigma$ being "an involution", that is

$$\begin{cases} \sigma(i_1) = i_1 \\ \sigma(i_2) = i_2 \end{cases} \quad \text{or} \quad \begin{cases} \sigma(i_1) = i_2 \\ \sigma(i_2) = i_1 \end{cases} \tag{1}$$

The identification accuracy is defined as

$$I := \frac{1}{N'} \sum_{(i_1, i_2)} \mathbf{1}_{\text{dist}(\boldsymbol{v}_{i_1}, \boldsymbol{f}_{i_1}) < \text{dist}(\boldsymbol{v}_{i_1}, \boldsymbol{f}_{i_2})}, \tag{2}$$

where $\text{dist}$ is a distance between representations (we take the cosine distance). This ratio represents the number of triplets where the audio's true representation is more similar to the representation of true face, compared to a random face from a different speaker. For the heterogeneous problem (Nagrani et al., 2018a) mention human's capacity in this metric is $81.3\%$, yet when considering the problem in a completely homogeneous domain (with variance in gender, age and nationality removed) the performance drops sharply to $57.1\%$. In addition to the identification accuracy it might be interesting to benchmark the system in *binary accuracy*, i.e., given $\frac{N'}{2}$ matching pairs $(\boldsymbol{v}_{i_1}, \boldsymbol{f}_{i_1})$ and $\frac{N'}{2}$ non-matching pairs $(\boldsymbol{v}_{i_1}, \boldsymbol{f}_{i_2})$ (with $i_1 \neq i_2$) how many of them is classified correctly. For that metric, we pick as a threshold for the binary classifier an $\epsilon$ yielding the equal error rate (EER).

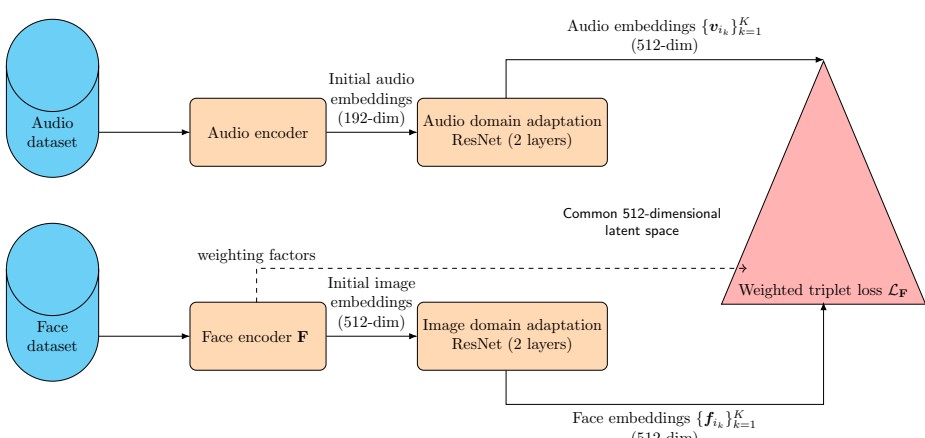

Figure 3: An outline of our model, relying on a Face Distance Weighted triplet loss

### 2.3.2 PERCENTILE RECALL

Assessing models performing voice face retrieval appears to be a more demanding task: naively one would attempt measuring voice face retrieval by examining a *"1:N'-accuracy"* (that is replace triplets used in subsubsection 2.3.1 with tuples of the form $(\boldsymbol{v}_{i_1}, \boldsymbol{f}_{\sigma(j_0)}...\boldsymbol{f}_{\sigma(j_{N'-1})})$ for some permutation of faces $\sigma$ attaining the value $i_1$ for some unique $j_k$. Nevertheless, as Nagrani et al. (2018a) mention the value of this extended matching is exponentially decaying as $N'$ increases. They explain it by the increasing probability of encountering face belonging to the same ethnicity, gender and age of the anchor speaker. Inspired by their argument we introduce a weaker yet informative metric, *percentile recall*: given an audio sample belonging to a speaker and a gallery of face images with one or more of the speaker's face, in which percentile is the correct face located *on average*? Roughly speaking that amounts for defining a continuous random variable $R$ which measures how well the model is able to place faces of the correct speaker in a large gallery (Refer to Figure 4 for the cumulative distributive functions we obtained from our experiments). We compute the metric using a Monte Carlo approach as follows:

1. **Setup:**

   (a) Consider a dataset of audio recordings belonging to $N$ speakers $\{s_1, \ldots, s_N\}$ and a corresponding dataset of their faces. Each speaker has $\tilde{a}_j$ corresponding audio samples in the audio dataset and $\tilde{p}_j$ face samples in the faces dataset with $\tilde{a}_j, \tilde{p}_j \geq 1$. By an abuse of notation, enumerating the faces and voices corresponding to a speaker $s_j$, we denote their $i-$th common latent face representation by $\mathrm{f}_i(s_j)$ and their $i-$th voice representation by $\mathrm{v}_i(s_j)$.

   (b) In addition fix large positive integers $M, a, p$ and $\tilde{n}$.

2. **Monte Carlo Simulation:** For each speaker $s_j$ ($j \in \{1, \ldots, N\}$) consider $a_j = \min(a, \tilde{a}_j)$ of their voice samples.

   (a) For each audio sample $\mathrm{v}_i(s_j)$:

      i. Sample at random $p_j = \min(p, \tilde{p}_j)$ "positive" face embeddings belonging to $s_j$.

      ii. Form a large gallery of negative face embeddings: pick another "negative" speaker $s_{j'}$, with $j' \neq j$, and append $n_{j'} = \min(\tilde{n}, \tilde{p}_{j'})$ face embeddings to a list of negative face embeddings. While the list has less than $\tilde{n}$ faces, repeat the process with negative speakers that weren't selected before. We end with a list of negative faces containing $n \geq \tilde{n}$ faces.

      iii. The positive and negative face images form a face gallery $G$ of size $n + p_j$.

      iv. Encode the audio recording and all gallery images using the cross-modal encoder.

      v. Compute cosine similarity between the audio embedding and each of the image embeddings.

vi. For each positive face embedding $f_{i'}(s_j)$, calculate its rank

$$\text{rank}(f_{i'}(s_j)) := \frac{\#\{\text{negative } n \in G : \cos(f_{i'}(s_j), v_i(s_j)) < \cos(n, v_i(s_j))\}}{n + p_j}$$

vii. The rank of an audio sample is then defined by averaging

$$\text{rank}(v_i(s_j)) := \frac{1}{p_j} \sum_{i'=1}^{p_j} \text{rank}(f_{i'}(s_j)),$$

viii. Append the audio rank to the list of all ranks.

(b) Repeat the simulation $M$ times.

3. **Statistical Analysis:** The continuous random variable $R$ is then approximated by the independent identically distributed random discrete variables $r_m$ (with $m \in \{1, \dots, M\}$) given by

$$P(r_m \leq \alpha) := \frac{\sum_{j=1}^{N} \#\{i \in \{1, \dots, a_j\} : \text{rank}(v_i(s_j)) \leq \alpha\}}{\sum_{j=1}^{N} a_j} \quad (3)$$

This Monte Carlo simulation provides a robust approximation of $R$, allowing for accurate computation of the percentile recall metric. It is often customary to compare between two models by comparing $p(R \leq \alpha)$ for fixed rational $\alpha$, attaining the form $\alpha = \frac{k}{N}$. This approximately describe the probability of a positive face appearing in top $k$ elements in a gallery of size $N$. For that purpose we define a table *Recall at N's* as we present in Table 2 , whose rows correspond to various gallery sizes $N'$, and whose columns to various $k < N'$ and elements of the table are given by $p(R \leq \frac{k}{N'})$.

## 3 THE MODEL

Our model is based on embedding voice and face embeddings of pre-processed media, as discussed in subsection 2.2, into a common latent space (see Figure 3) while teaching the model robust and discriminative features for identity recognition. We begin with a pair of encoders generating embeddings for each modality separately: for audio samples we use TitaNet embedding (Koluguri et al., 2022) and for face images a IR-SE50, combination of IR-50 (He et al., 2016) with SENet (Hu et al., 2018), pre-trained with ArcFace loss (Deng et al., 2019). In addition the architecture consists of two feed-forward neural networks outputting vectors of size 512. This pair of domain adaptation networks is trained using a unique loss function. We apply a version of triplet loss due to Ivanov & Krishtul (2023) called *face distance weighted triplet loss*: fixing $f : \mathbb{R} \to \mathbb{R}$ be a non-decreasing function (i.e., sigmoid), we damp the summands by the *initial representations distance*,

$$\mathcal{L}_{\mathbf{F}} := \sum_{k=1}^{K} \left( ||\boldsymbol{v}_{i_k} - \boldsymbol{f}_{i_k}||^2 - ||\boldsymbol{v}_{i_k} - \boldsymbol{f}_{j_k}||^2 + \beta \right)^{+} \cdot f(\text{dist}(\boldsymbol{f}_{i_k}^{\mathbf{F}}, \boldsymbol{f}_{j_k}^{\mathbf{F}})) \quad (4)$$

where

- $\beta$ is the triplet loss margin constant
- the summation is done over the $K$ triplets in the batch.
- for every $k \in \{1, \dots, K\}$, $i_k$ is the $k-$th anchor speaker in the batch and $j_k \neq i_k$ is a different speaker.
- $\boldsymbol{f}_{i_k}^{\mathbf{F}}, \boldsymbol{f}_{j_k}^{\mathbf{F}}$ are the representations obtained from a frozen pre-trained face recognition network $\mathbf{F}$ (in our settings, a IR-SE50 network trained with ArcFace loss) of the $i_k-$th and $j_k-$th speakers respectively.
- dist is a distance between these representations. We consider cosine distance.

Compared to the standard triplet loss method employed for instance in FaceNet Schroff et al. (2015), this damping process offers enhanced biometric performance by increasing the penalty for errors between dissimilar facial features while reducing it for mis-identifications for similar-looking speakers.

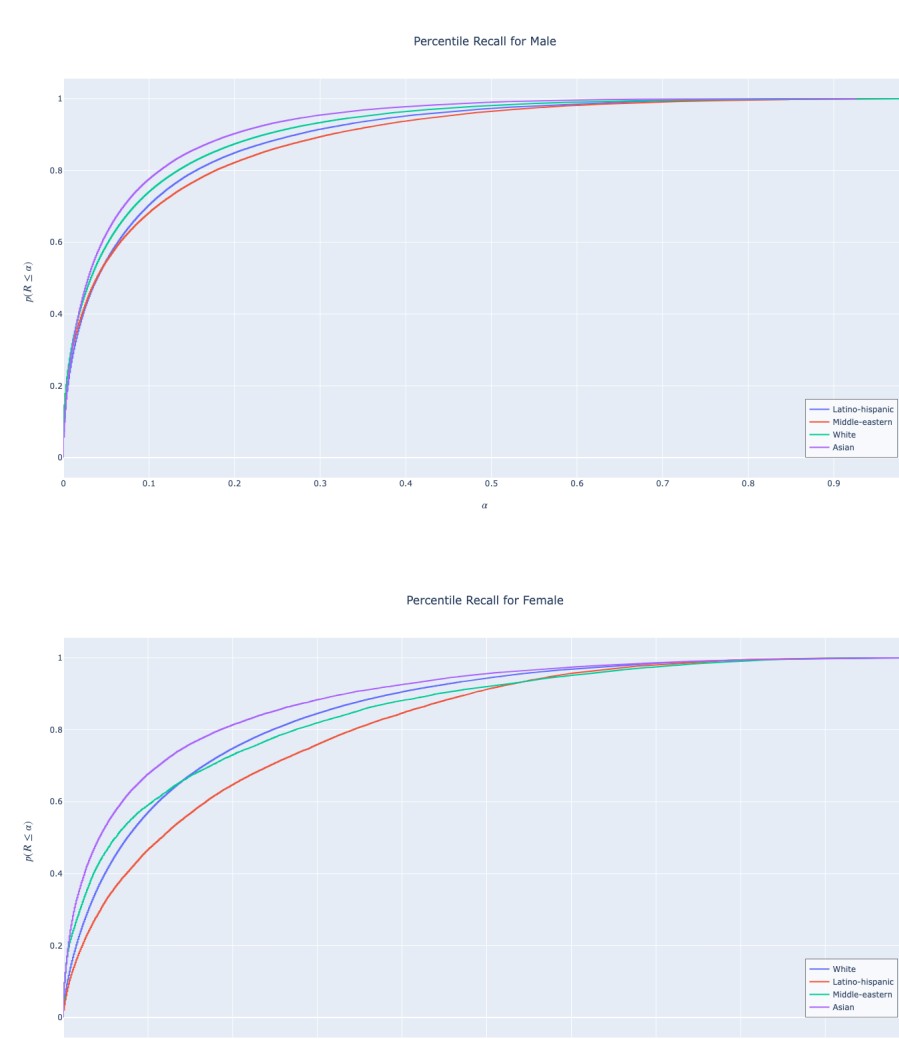

Figure 4: The cumulative distribution functions of the percentile recall: (top) masculine homogeneous population and (bottom) feminine homogeneous population

# 4 RESULTS

Our experiment consists of two stages. First we pre-train a heterogeneous model, i.e., on the demographically-varied datasets from subsubsection 2.1.1. We then fine-tune this model on each of the homogeneous populations sharing gender and ethnicity. We use both for pre-training and for fine-tuning the same architecture and weighted triplet loss mentioned above.

Our models' performance in traditional biometric metrics is given in Figure 5a. Comparing the 1:N accuracy Figure 5b) and Figure 5c we observe a consistent performance hierarchy across demographic groups as gallery size increases. Asian males and females maintain the highest accuracy within their respective genders, while Latino-Hispanic males and White females show the steepest degradation with increasing N. The performance gap between genders widens at larger gallery sizes, with female groups showing more pronounced accuracy drops.

Examining these patterns in detail, , our voice-face matching system demonstrates varying performance across demographic groups, with patterns suggesting complex interactions beyond simple data quantity effects. While the system achieves the highest binary matching accuracy of 85.55% for Asian males, followed by White males at 83.99%, the performance variations between groups

| top
out of | 5 | 10 | 50 | 100 | 250 | 500 | 1000 | 5000 |
|---|---|---|---|---|---|---|---|---|
| 100 | 0.592 | 0.741 | 0.981 | 1 | – | – | – | – |
| 1000 | 0.244 | 0.318 | 0.592 | 0.741 | 0.909 | 0.981 | 1 | – |
| 10,000 | 0.101 | 0.147 | 0.244 | 0.318 | 0.456 | 0.592 | 0.741 | 0.981 |

Table 2: Recall at N' with various N's and ks: probability of correct face retrieved in top k places out of gallery of size N'. Obtained by sampling the precision recall curve of the model fine-tuned on white male population.

point to the importance of intra-group diversity. We observe a consistent gender gap across all demographic groups, with male subjects achieving higher performance across all three metrics (binary accuracy, AUC, and identification accuracy). For instance, Asian females achieve 80.78% binary accuracy compared to 85.55% for Asian males. This performance gap persists even in demographic groups with substantial training data, suggesting that the diversity of features within each demographic group may be as crucial as the raw quantity of training examples. For the voice-face retrieval task, Figure 4 presents the cumulative distribution functions stratified by gender across demographic groups. Notably, while expanding the dataset improves performance, this improvement specifically stems from increased intra-group variance - maintaining fixed gender and ethnicity while diversifying other attributes such as age and facial features.

As an alternative performance measure, we evaluated our system using the Recall at N' metric defined in subsubsection 2.3.2, which quantifies how often the correct match appears within a given gallery percentile. Table 2 demonstrates this analysis for the White male population, showing the model positions correct faces within the top 10 percentile with probability 0.741.

## 5 PERSPECTIVES

Our analysis of voice-face matching in demographically homogeneous settings reveals both methodological insights and crucial data challenges. While our results demonstrate the system's capability to learn cross-modal associations within demographic groups, they also highlight the complex role of intra-group variance - suggesting that attributes like age and facial features significantly influence matching performance even when gender and ethnicity are fixed. This points to two parallel imperatives for advancing the field: expanding data collection for currently underrepresented groups (particularly Black and Indian populations), while simultaneously investigating how specific voice and face attributes impact matching performance within demographic groups. Such dual focus would enable both broader demographic coverage and deeper understanding of which multimodal features drive successful matching across different populations. Future work could systematically vary additional covariates (e.g., age groups, accent variations, facial characteristics) within demographically homogeneous groups to isolate their impact on cross-modal learning, providing insights into the robustness and fairness of voice-face matching systems.

### REPRODUCIBILITY OF RESULTS

Our results can be reproduced using publicly available components: The datasets shown in Figure 2 are available upon request from the respective research teams. Our homogeneous datasets can be curated by applying DeepFace Serengil & Ozpinar (2021) demographic classification on AVSpeech. Both TitaNet and IR-SE50 face encoder weights are publicly available. During pre-training, our weighted sampler selected identities with probability 0.5 from AVSpeech and 0.25 from each of the other datasets. The inter-domain networks use dropout= 0.2. Models were trained on NVIDIA RTX 3070 and 3080 GPUs.

### REFERENCES

Sanyuan Chen, Chengyi Wang, Zhengyang Chen, Yu Wu, Shujie Liu, Zhuo Chen, Jinyu Li, Naoyuki Kanda, Takuya Yoshioka, Xiong Xiao, et al. Wavlm: Large-scale self-supervised pre-training for

| Gender | Ethnicity | Bin. ACC (1:1) | AUC | Iden. Acc (1:2) |
|---|---|---|---|---|
| Male | Asian | 85.55 | 92.71 | 92.39 |
| | Latino-Hispanic | 82.94 | 90.64 | 90.29 |
| | Middle-Eastern | 81.77 | 89.96 | 89.3 |
| | White | 83.99 | 91.9 | 91.43 |
| Female | Asian | 80.78 | 89.15 | 88.41 |
| | Latino-Hispanic | 73.03 | 81.93 | 81.83 |
| | Middle-Eastern | 77.15 | 85.33 | 86.52 |
| | White | 77.74 | 85.87 | 85.7 |

(a) Performance metrics across gender and ethnicity groups showing Binary Accuracy (1:1), AUC, and Identification Accuracy (1:2).

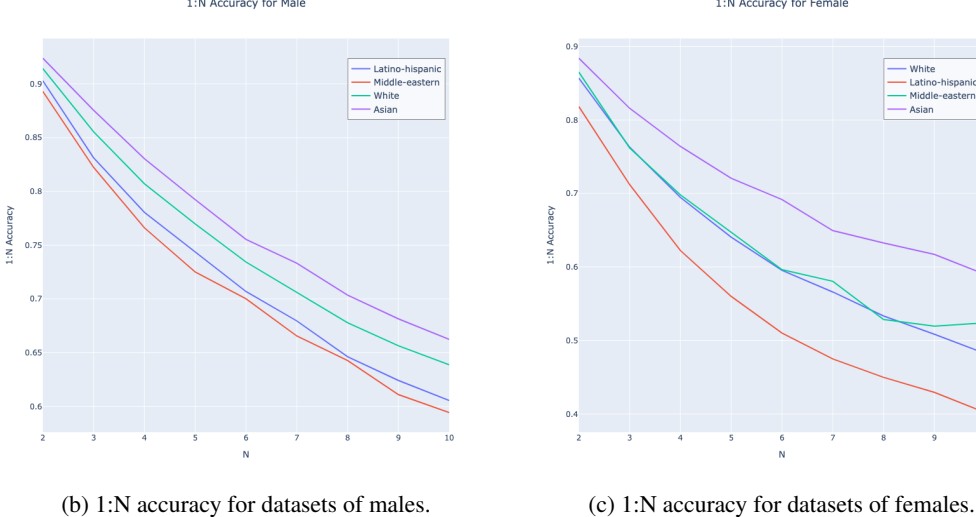

(b) 1:N accuracy for datasets of males.       (c) 1:N accuracy for datasets of females.

Figure 5: Performance analysis across gender and ethnicity groups.

full stack speech processing. *IEEE Journal of Selected Topics in Signal Processing*, 16(6):1505–1518, 2022a.

Sanyuan Chen, Yu Wu, Chengyi Wang, Zhengyang Chen, Zhuo Chen, Shujie Liu, Jian Wu, Yao Qian, Furu Wei, Jinyu Li, et al. Unispeech-sat: Universal speech representation learning with speaker aware pre-training. In *ICASSP 2022-2022 IEEE International Conference on Acoustics, Speech and Signal Processing (ICASSP)*, pp. 6152–6156. IEEE, 2022b.

Jiankang Deng, Jia Guo, Niannan Xue, and Stefanos Zafeiriou. Arcface: Additive angular margin loss for deep face recognition. In *Proceedings of the IEEE/CVF conference on computer vision and pattern recognition*, pp. 4690–4699, 2019.

Ariel Ephrat, Inbar Mosseri, Oran Lang, Tali Dekel, Kevin Wilson, Avinatan Hassidim, William T Freeman, and Michael Rubinstein. Looking to listen at the cocktail party: A speaker-independent audio-visual model for speech separation. *ACM transactions on Graphics*, 37(4):1–11, 2018.

Kaiming He, Xiangyu Zhang, Shaoqing Ren, and Jian Sun. Deep residual learning for image recognition. In *Proceedings of the IEEE conference on computer vision and pattern recognition*, pp. 770–778, 2016.

Jie Hu, Li Shen, and Gang Sun. Squeeze-and-excitation networks. In *Proceedings of the IEEE conference on computer vision and pattern recognition*, pp. 7132–7141, 2018.

Danil Ivanov and Arkady Krishtul. System and method for matching a voice sample to a facial image, 2023. URL https://patents.justia.com/patent/11847804.

Minchul Kim, Anil K Jain, and Xiaoming Liu. Adaface: Quality adaptive margin for face recognition. In *Proceedings of the IEEE/CVF conference on computer vision and pattern recognition*, pp. 18750–18759, 2022.

Minchul Kim, Yiyang Su, Feng Liu, Anil Jain, and Xiaoming Liu. Keypoint relative position encoding for face recognition. In *Proceedings of the IEEE/CVF Conference on Computer Vision and Pattern Recognition*, pp. 244–255, 2024.

Nithin Rao Koluguri, Taejin Park, and Boris Ginsburg. Titanet: Neural model for speaker representation with 1d depth-wise separable convolutions and global context. In *ICASSP 2022-2022 IEEE International Conference on Acoustics, Speech and Signal Processing (ICASSP)*, pp. 8102–8106. IEEE, 2022.

Lauren W Mavica and Elan Barenholtz. Matching voice and face identity from static images. *Journal of Experimental Psychology: Human Perception and Performance*, 39(2):307, 2013.

Arsha Nagrani, Samuel Albanie, and Andrew Zisserman. Seeing voices and hearing faces: Cross-modal biometric matching. In *Proceedings of the IEEE conference on computer vision and pattern recognition*, pp. 8427–8436, 2018a.

Arsha Nagrani, Samuel Albanie, and Andrew Zisserman. Learnable pins: Cross-modal embeddings for person identity. In *Proceedings of the European Conference on Computer Vision (ECCV)*, pp. 71–88, 2018b.

Arsha Nagrani, Joon Son Chung, Weidi Xie, and Andrew Zisserman. Voxceleb: Large-scale speaker verification in the wild. *Computer Science and Language*, 2019.

Tae-Hyun Oh, Tali Dekel, Changil Kim, Inbar Mosseri, William T Freeman, Michael Rubinstein, and Wojciech Matusik. Speech2face: Learning the face behind a voice. In *Proceedings of the IEEE/CVF conference on computer vision and pattern recognition*, pp. 7539–7548, 2019.

Bilal Porgali, Vítor Albiero, Jordan Ryda, Cristian Canton Ferrer, and Caner Hazirbas. The casual conversations v2 dataset. In *Proceedings of the IEEE/CVF Conference on Computer Vision and Pattern Recognition*, pp. 10–17, 2023.

Florian Schroff, Dmitry Kalenichenko, and James Philbin. Facenet: A unified embedding for face recognition and clustering. In *Proceedings of the IEEE conference on computer vision and pattern recognition*, pp. 815–823, 2015.

Sefik Ilkin Serengil and Alper Ozpinar. Hyperextended lightface: A facial attribute analysis framework. In *2021 International Conference on Engineering and Emerging Technologies (ICEET)*, pp. 1–4. IEEE, 2021. doi: 10.1109/ICEET53442.2021.9659697. URL https://ieeexplore.ieee.org/document/9659697/.

Harriet MJ Smith, Andrew K Dunn, Thom Baguley, and Paula C Stacey. Concordant cues in faces and voices: Testing the backup signal hypothesis. *Evolutionary Psychology*, 14(1): 1474704916630317, 2016a.

Harriet MJ Smith, Andrew K Dunn, Thom Baguley, and Paula C Stacey. Matching novel face and voice identity using static and dynamic facial images. *Attention, Perception, & Psychophysics*, 78:868–879, 2016b.

Timothy Wells, Thom Baguley, Mark Sergeant, and Andrew Dunn. Perceptions of human attractiveness comprising face and voice cues. *Archives of sexual behavior*, 42:805–811, 2013.

Yandong Wen, Mahmoud Al Ismail, Weiyang Liu, Bhiksha Raj, and Rita Singh. Disjoint mapping network for cross-modal matching of voices and faces. In *International Conference on Learning Representations*, 2019. URL https://openreview.net/forum?id=B1exrnCcF7.

Kaipeng Zhang, Zhanpeng Zhang, Zhifeng Li, and Yu Qiao. Joint face detection and alignment using multitask cascaded convolutional networks. *IEEE signal processing letters*, 23(10):1499–1503, 2016.

Boqing Zhu, Kele Xu, Changjian Wang, Zheng Qin, Tao Sun, Huaimin Wang, and Yuxing Peng. Unsupervised voice-face representation learning by cross-modal prototype contrast. *arXiv preprint arXiv:2204.14057*, 2022.

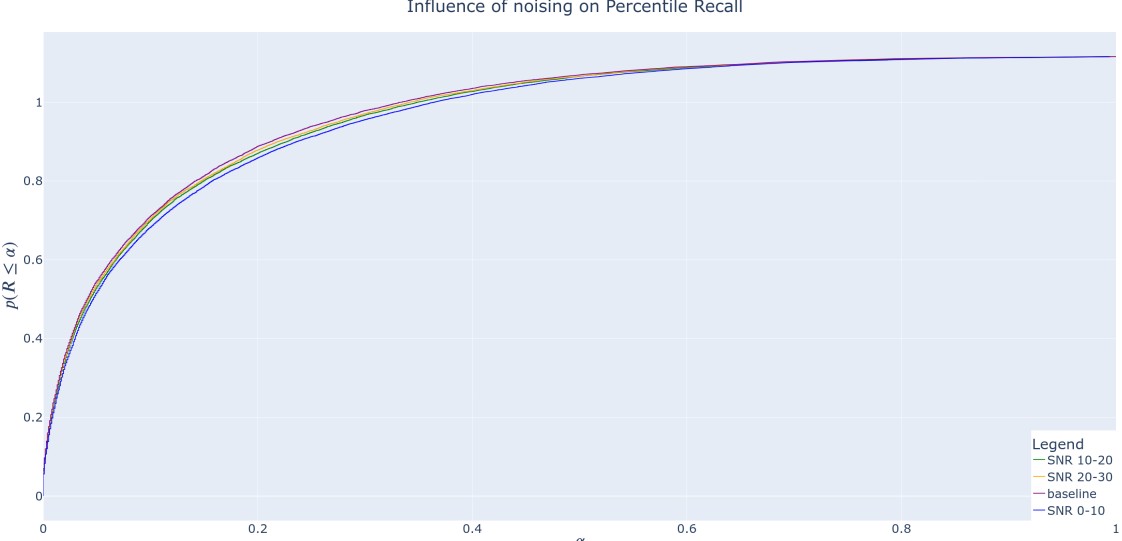

Figure 6: The CDFs obtained from noising the audios

# A  SENSITIVITY TO VARIOUS AUDIO CONDITIONS ON WHITE MALE POPULATION

Our model whose KPIs were given in Table 2 was only trained on data gathered from YouTube, which differs significantly from data encountered in mundane situations. Therefore, we assess its performance on challenging settings. We examine the situation only for the model trained for white male population, for which there are statistically evident quantity of people in order to show the effect of such settings on our model.

## A.1  INFLUENCE OF NOISE INJECTION

In this test we noised our audios of white males from AVSpeech and re-evaluated our performance in the percentile recall metric. Noising is measured in terms of the Signal-to-noise ratio (SNR): the higher it is the noisier the signal is. We consider three scenarios: of minor (20-30 SNR), medium (10-20 SNR) and major noising (0-10 SNR). We draw the respective CDFs in Figure 6 and notice that our model shows minor sensitivity for noise, both when examining minor and of major noising. In the latter we observe a decrease of 4% in terms of being in top 10 percentile.

This sensitivity to noise can be improved by introducing random noising as an augmentation performed during training.

## A.2  INFLUENCE OF CODECS

Another interesting application touches the compression and encoding of audios through modern communication devices, like telephones and mobile devices. We passed the audio recordings through 4 different codecs:

- AMR (Adaptive Multi-Rate): A speech compression codec designed for mobile networks. It supports multiple bit rates ranging from 4.75 to 12.2 kbps and adapts to network conditions for optimal quality.

- G729: A low-bandwidth speech codec widely used in VoIP applications, operating at 8 kbps. It offers a good balance between audio quality and bandwidth efficiency, making it popular for internet telephony.

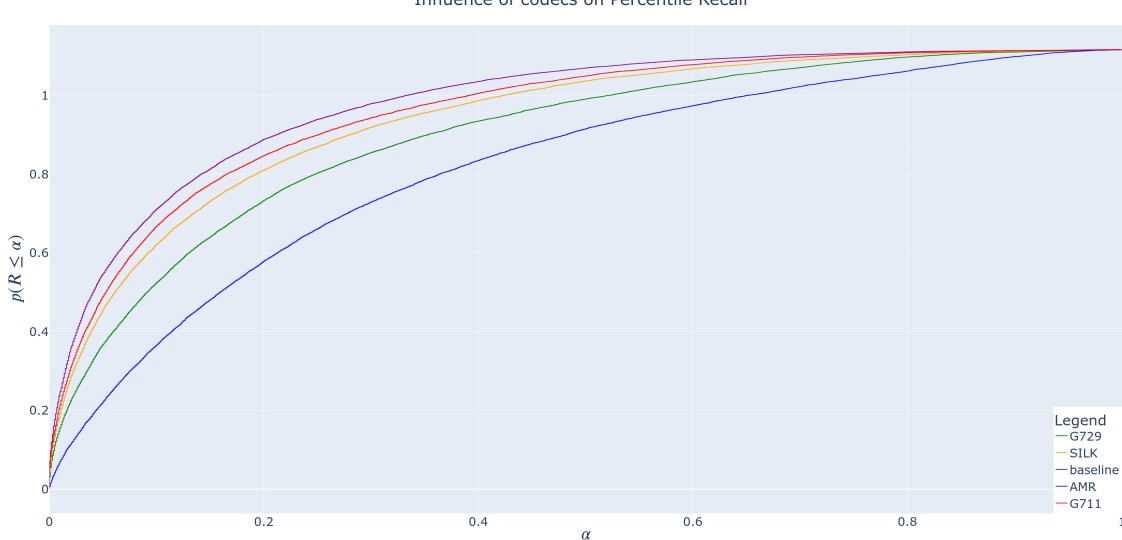

Figure 8: Precision recall curves obtained from adding codecs to the original audio recordings. AMR and G729 cause significant drop in performance

- SILK: SILK is a variable bit rate codec, developed by Skype, operating between 6 to 40 kbps. It's optimized for both speech and music transmission, providing flexibility for various audio content types.
- G711: The standard codec for telephone systems and VoIP, using pulse code modulation (PCM) at 64 kbps. It provides high-quality audio with minimal processing delay, making it ideal for scenarios where bandwidth is not a constraint.

We then calculated the percentile-recall metric for each dataset of encoded audios (see the results in Figure 8). In SILK and G711 we observe minor drop in performance yet when applying G729 and AMR we experience major drop of ranks. We conjecture this phenomenon is due to the "aggressive" nature of their compression . We suggest adding to training procedure codecs as part of artificial augmentation of the data, with high percentage of audios being encoded using the more challenging codecs, AMR and G729.

### A.3 COMBINED
INFLUENCE OF CODECS AND NOISING

We ran additional trials combining each of the codecs we mention above with minor, intermediate and major noising as before. We observe a major decrease in the performance of model in all scenarios. We refer the reader to Figure 9,Figure 10,Figure 11,Figure 12 for the full CDFs. Finally we refer the reader to Figure 7 for the drop in the identification accuracy in the examined scenarios.

| Codec | SNR | ID Acc. (%) |
|-------|-----|-------------|
| None  | -     | 91.43 |
|       | 20-30 | 91.15 |
|       | 10-20 | 90.73 |
|       | 0-10  | 89.71 |
| AMR   | -     | 74.16 |
|       | 20-30 | 73.73 |
|       | 10-20 | 72.62 |
|       | 0-10  | 68.14 |
| SILK  | -     | 87.42 |
|       | 20-30 | 86.06 |
|       | 10-20 | 85.55 |
|       | 0-10  | 83.49 |
| G729  | -     | 83.03 |
|       | 20-30 | 81.90 |
|       | 10-20 | 81.16 |
|       | 0-10  | 77.00 |
| G711  | -     | 88.49 |
|       | 20-30 | 89.14 |
|       | 10-20 | 87.32 |
|       | 0-10  | 87.08 |

Figure 7: Identification Accuracy for Various Codecs and SNR Ranges (White male population).

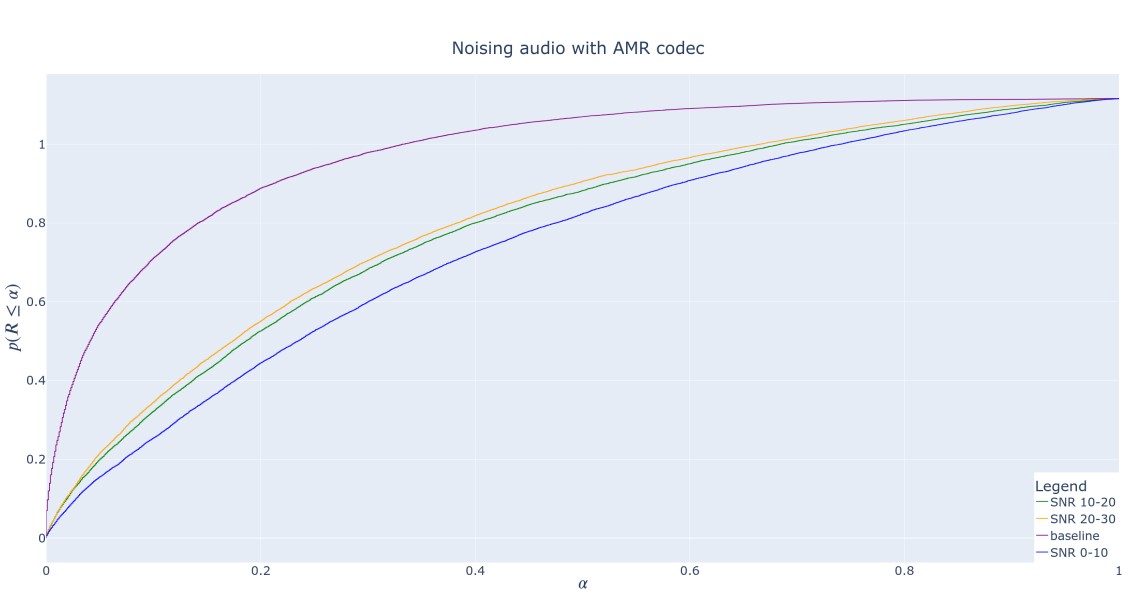

Figure 9: Percentile recall curves obtained from noising AMR encoded audios

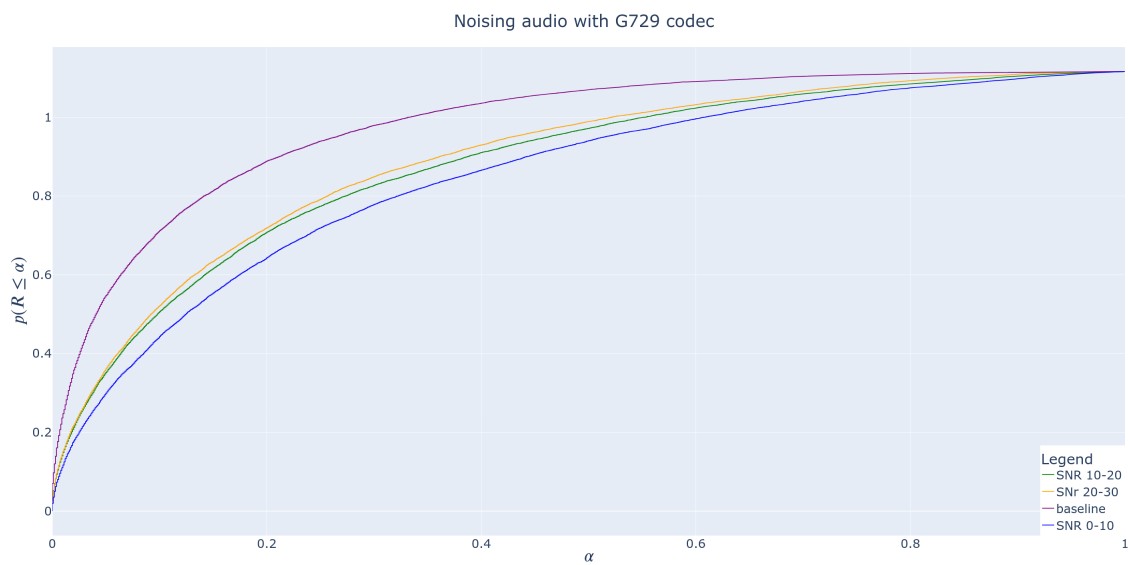

Figure 10: Percentile recall curves obtained from noising G729 encoded audios

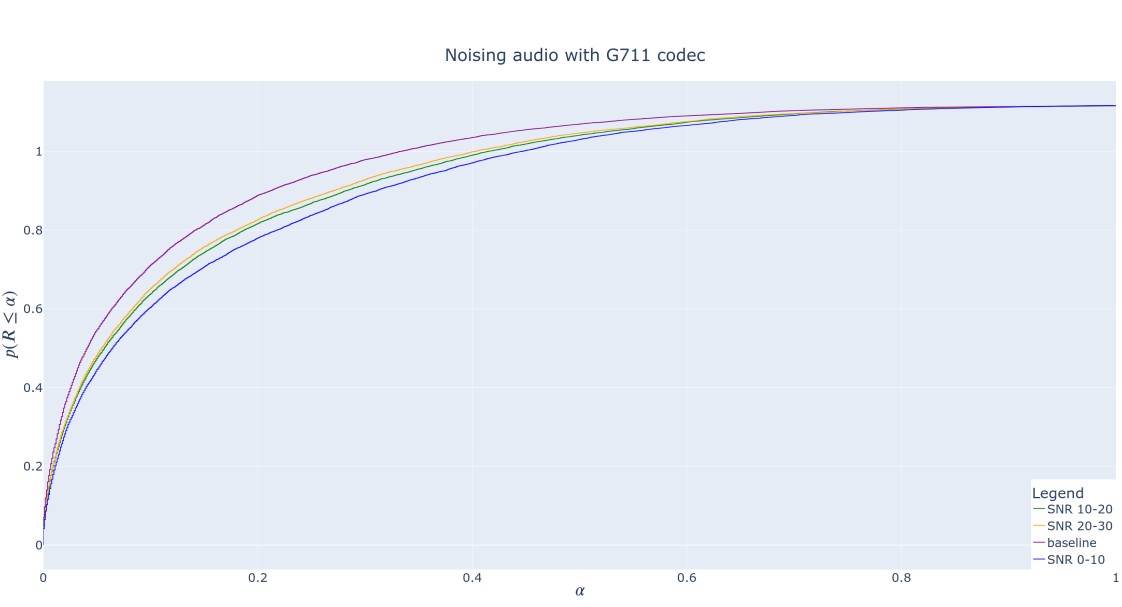

Figure 11: Percentile recall curves obtained from noising G711 encoded audios

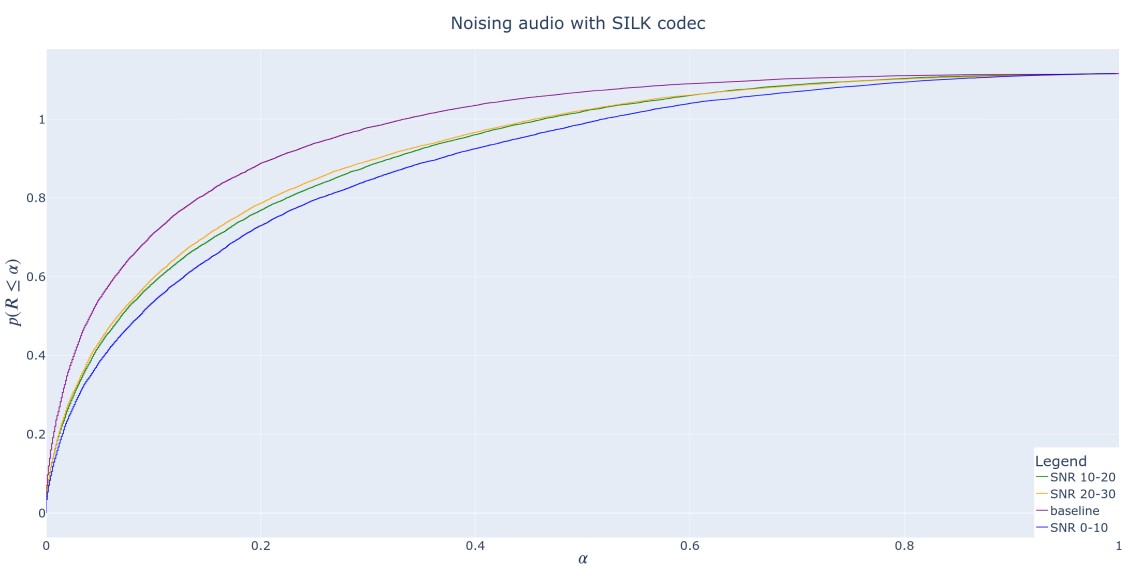

Figure 12: Percentile recall curves obtained from noising SILK encoded audios

