# OpenReview forum: "Hearing faces among homogeneous populations: improvement of cross-modal biometrics"
_ICLR.cc/2025/Conference — Submitted to ICLR 2025_

### Official Review · Reviewer_uBYY · 2024-11-03

**Soundness:** 2
**Presentation:** 2
**Contribution:** 2
**Rating:** 3
**Confidence:** 4

**Summary:**

The authors make improvements to the task of voice face matching (and retrieval). The authors claim that existing approaches are limited because their datasets do not have homogeneity in the population, which makes the task easier. In order to solve the task, the authors introduce a new dataset which contains samples of population which are similar in demographic characteristics. They also develop a new deep learning architecture which is claimed to attain better results than the existing methods.

The key contributions of the paper:
- Introduction of the homogeneous dataset.
- A novel architecture which the authors claim works better than existing methods.
- Introduction of the percentile recall metric for voice face retrieval.

**Strengths:**

1. The authors identify a weak point in existing literature (high variance makes task easier) and attempt to tackle it.
2. The equations for loss and the newly introduced metric are clearly illustrated.
3. Discussion on the influence of noise injection is factored in.
4. The authors introduce a new metric (percentile recall) which tackles the problem of exponential decay in existing metrics in voice face retrieval.
5. Section on reproducibility is included.

**Weaknesses:**

1. Table 2 in the results section is not very clear. The authors mention for table 2 that "1:2 accuracy on homogeneous data in present work compared to previous works." It is unclear what is meant by "homogeneous data". It appears that the metrics being reported are for the methods on different GNA-var removed datasets and NOT each method on the dataset that the authors introduced.
2. If the above is in fact true, then this seems like a problem – different methods applied on different datasets cannot lead to a fair assessment of the proposed method. What is ideally needed, is the metrics of existing methods on the dataset that the authors propose for comparison.
3. The authors correctly state that "As Nagrani et al. (2018a) note, human performance deteriorates markedly when assessed on voice face matching of speakers sharing gender, ethnicity, or age group." However, Nagrani et al. (2018a) also report metrics (which I think are provided in table 2) with GNA-var removed. Now, I am unclear on the value the homogenous dataset introduced adds? Is it the size?
4. If the answer to 3 is yes, then the key finding appears to be that training on more "relevant" data (or hard examples) is helpful. This itself as a contribution to the field seems weak in my opinion.
5. The authors only target one demographic. There is no discussion on why this demographic was chosen. I believe it would be important for proving the efficacy by applying the method on different groups individually. I also believe that it would be interesting to see the resulting patterns as well as reduce bias.
6. The network architecture diagram (Fig. 3) is very high level, and it could benefit with the addition of more details like layers, parameters, etc.
7. Lacks implementation details like learning rate, number of layers, etc.

**Questions:**

1. It is not clear to me why audio is split into 3-8 seconds intervals? Why is this variable? How is the decision made?
2. N' is not clearly defined in section 2.3.1.
3. The authors write for the dataset – "The list of them is available upon request". It is not clear if this paper has a dataset contribution.
4. More information on how the data was collected (or more importantly filtered) is lacking.

---

> ### Author Response · Authors · 2024-12-01
>
> Dear Reviewer uBYY03,
>
> We wish to thank you for your detailed response concerning our work. We have uploaded a revised version with much wider scope, addressing the performance of our model on various homogeneous populations extracted from the AVSpeech dataset. We benchmarked each of the models using additional metrics, as reviewer eoJD proposed.
>
> Regarding the described weaknesses:
>
> 1+2. Accepted. Previous works were trained on other homogeneous populations, inducing additional bias to the comparison. As a solution, we now supply a table of metrics on various homogeneous populations extracted from AVSpeech.
>
> 3+4. We removed White Robin from consideration. We discuss quantity of data vs. results in various metrics in the revised results section.
>
> 5. Added 3 more demographics and an additional gender, bringing the number of examined populations to (1+1)×(1+3)=8.
>
> 6. Improved the network architecture figure.
>
> 7. Added additional implementation details.
>
> Regarding the questions:
>
> 1. Most audios in VoxCeleb are at most 8 seconds long. We chose 3 seconds as the minimal length to allow the audio encoder to extract a meaningful representation of the speaker. When preprocessing the data, we trim the audio uniformly to be 3-8 seconds long.
>
> 2. Replaced White Robin with subsets of publicly available dataset AVSpeech.
>
> 3. We believe this question concerns White Robin, which is no longer present in this version of the paper. The data collection practices of the other datasets are available in their respective works.

---

### Official Review · Reviewer_eoJD · 2024-11-04

**Soundness:** 1
**Presentation:** 1
**Contribution:** 1
**Rating:** 3
**Confidence:** 5

**Summary:**

The paper makes significant contributions to the field of cross-modal biometrics by proposing a novel deep architecture and a new evaluation metric tailored for homogeneous datasets. The weighted triplet loss function effectively improves the model's performance in challenging scenarios. However, the model's generalizability and practicality need further investigation.

**Strengths:**

This paper introduces several key innovations aimed at improving voice-face matching in a homogeneous dataset. The primary innovation lies in a homogeneous dataset White Robin, where speakers share gender and ethnicity. And then propose a deep architecture that leverages a weighted triplet loss function based on face distances. To better evaluate the performance of voice-face retrieval tasks, the authors propose a new metric called percentile-recall.

**Weaknesses:**

1. The innovation in this article is insufficient. (1) The proposed weighted triplet loss is very similar to the triplet loss
used in FaceNet [1]. (2) The mechanism of the introduced Percentile-Recall is closely related to existing retrieval
performance evaluation methods.

2. The experiments are not sufficiently comprehensive to fully support the work. How does the proposed method
perform on general metrics for voice-face matching, such as Binary Accuracy (ACC), Multi-way Accuracy (ACC),
and Verification Area Under the Curve (AUC) [2]？

[1] Schroff F, Kalenichenko D, Philbin J. Facenet: A unified embedding for face recognition and clustering[C]//
Proceedings of the IEEE conference on computer vision and pattern recognition. 2015: 815-823.
[2] Wang J, Zheng A, Yan Y, et al. Attribute-guided Cross-modal Interaction and Enhancement for Audio-Visual
Matching[J]. IEEE Transactions on Information Forensics and Security, 2024.

**Questions:**

Why is it necessary to propose a homogeneous dataset?
What is the correlation between human appearances and voices within the same gender and ethnicity？

**Details Of Ethics Concerns:**

The dataset used in this study was collected from YouTube. Will the dataset be made public? Are there privacy and copyright issues?

---

> ### Author Response · Authors · 2024-12-01
>
> Dear Reviewer eoJD,
> Thank you for your response concerning our work. We have taken active measures to improve its quality in the current version:
>
> 1. As you suggested, we have replaced the White Robin dataset with models trained on 8 homogeneous groups (2 genders × 4 ethnicities) gathered from AVSpeech. We trained a model on each of these groups and benchmarked it using the metrics you suggested.
>
> 2. Regarding the loss function: While FaceNet uses "vanilla" triplet loss that assigns equal weight to all mistakes where the anchor embedding was closer to the negative than to the positive embedding, our loss assigns different weights to each such triplet. This weighting is crucial in homogeneous datasets - for example, confusing an elderly Asian woman with a young adult one is not as severe as confusing her with another elderly Asian woman. This distinction helps the model learn finer-grained discriminative features.
>
> 3. Concerning percentile-recall: Traditional retrieval tasks typically consider only the first matching image, not the entire set. Our metric averages over all matches, reducing evaluation bias from outliers. For instance, if most images have low similarity while one has almost perfect similarity, our percentile-recall would indicate that the model is not properly handling the use-case.
>
> We believe these changes and clarifications provide a more complete description of our contribution to the field.

---

### Official Review · Reviewer_haHz · 2024-11-04

**Soundness:** 3
**Presentation:** 3
**Contribution:** 4
**Rating:** 5
**Confidence:** 4

**Summary:**

The paper investigates the problem of cross-modal biometric matching—specifically, associating voices with faces—within homogeneous datasets, focusing on populations sharing specific demographic traits, such as gender and ethnicity. The authors introduce a new deep architecture incorporating a face-distance-weighted triplet loss to optimize matching between faces and voices in datasets with reduced inter-speaker variability. They also propose “percentile recall” as a novel metric to evaluate voice-to-face retrieval accuracy in large galleries. Their experiments demonstrate that the approach achieves state-of-the-art results with their new model architecture and fine-tuning of homogeneous data, surpassing existing models in voice-face matching accuracy.

**Strengths:**

- **Originality and Novelty**: The focus on homogeneous datasets for voice-face matching is a valuable addition to cross-modal biometrics research, addressing a specific gap in the field.
- **Metric Innovation**:  The introduction of percentile recall provides a practical metric for real-world applications where retrieval from large datasets is necessary.
- **State-of-the-Art Results**:  The paper reports superior performance using their proposed model and metric, setting a new benchmark for cross-modal matching in homogeneous datasets.
- **Detailed Experimental Setup**:  Extensive testing with various noise levels and codec conditions shows the model’s robustness, a critical aspect for practical deployment.

**Weaknesses:**

- **Limited Generalizability Discussion**: While the homogeneous dataset approach is compelling, the paper could further discuss its potential limitations in generalizing across other homogeneous populations (e.g., different ethnic groups).
- **Sensitivity to Loss Hyperparameters**: The performance impact of different parameter settings, particularly for the triplet loss function, is briefly mentioned but not thoroughly analyzed. A deeper analysis could clarify its robustness.
- **Comparative Evaluation**: More detailed comparative analysis with traditional metrics (besides identification accuracy) across various heterogeneous models would provide more precise insights into the model’s unique contributions.
- **Experiments**: More benchmarks, ablation studies, and result analysis (as mentioned above) could yield a more complete story. Improved figures and layout would help (use figures/plots to help tell the story and make them compelling and full of information, using space efficiently).

**Questions:**

1. **Generalizability of Model Across Different Homogeneous Groups**: How would the model adapt if trained on homogeneous datasets featuring different demographics (e.g., age or ethnicity)?

2. **Impact of Triplet Loss Weights**: Could you elaborate on the sensitivity of the weighted triplet loss function to variations in the face embedding space, especially for similar-looking individuals?

3. **Dataset Bias**: Given the heavy reliance on a particular homogeneous dataset (White Robin), how might this affect the model’s adaptability or performance on non-represented populations?

**Details Of Ethics Concerns:**

Based on the paper’s current content, no ethics review is needed. However, since it focuses on a single demographic (white males), ensuring diversity in dataset selection could be an ethical consideration for future work.

---

> ### Author Response · Authors · 2024-12-01
>
> Dear Reviewer haHz,
>
> We wish to thank you for your detailed review. We have uploaded an extended version of the original work. The major change with respect to the previous version is that we perform similar experiments on various populations and genders, removing the bias induced by the White Robin dataset. We have added additional biometric metrics (1:1 accuracy, 1:N accuracy, AUC) for each model, allowing us to obtain a more complete picture concerning the method and its efficiency on various homogenous populations. These metrics now appear in Figure 5 in the results section.
>
> We believe our contribution in its current state addresses questions 1 and 3 and most weaknesses. Regarding question 2: We think to address it, one must examine our model on "more homogenous data" (that is, people sharing not only gender and ethnicity but additional features like facial features or age). Gathering such highly homogenous data is a challenging task on its own, requires collection of mass amounts of labeled data, and may be considered a separate significant contribution.

---

### Meta-Review · Area_Chair_gN2b · 2024-12-17

**Metareview:**

The paper investigates voice-face matching on homogeneous datasets, introducing a new deep architecture, a face-distance-weighted triplet loss and a novel percentile recall metric to improve retrieval accuracy. The advantages of this paper include addressing a relevant gap in cross-modal biometrics, introducing a practical evaluation metric, and demonstrating robust results under noisy conditions. However, there are several weaknesses of this paper. First, the novelty of the loss function is limited, comparative experiments with standard metrics are insufficient. Additionally, the work lacks generalizability across demographics, thorough implementation details, and addresses dataset bias inadequately. Given the unclear significance of the contributions, I recommend rejecting the paper.

**Additional Comments On Reviewer Discussion:**

During the rebuttal, reviewers raised concerns about dataset bias, insufficient generalizability, unclear contributions of the loss function, and limited experimental scope. The authors addressed these by replacing the White Robin dataset with AVSpeech subsets covering 8 homogeneous groups (2 genders × 4 ethnicities), adding more biometric metrics (1:1 accuracy, 1:N accuracy, AUC), and clarifying the weighted triplet loss's role in homogeneous settings. They also expanded implementation details and improved visualizations. While these revisions improve robustness and address dataset concerns, the incremental novelty and lingering questions on broader generalizability led to the decision to reject the paper.

---

### Decision · Program_Chairs · 2025-01-22

Reject